# Differences in Help-Seeking Behavior among University Students during the COVID-19 Pandemic Depending on Mental Health Status: Results from a Cross-Sectional Survey

**DOI:** 10.3390/bs13110885

**Published:** 2023-10-25

**Authors:** Lukas Guenthner, Sabrina Baldofski, Elisabeth Kohls, Jan Schuhr, Tanja Brock, Christine Rummel-Kluge

**Affiliations:** 1Department of Psychiatry and Psychotherapy, Medical Faculty, Leipzig University, 04103 Leipzig, Germany; lukas.guenthner@medizin.uni-leipzig.de (L.G.); sabrina.baldofski@medizin.uni-leipzig.de (S.B.); elisabeth.kohls@medizin.uni-leipzig.de (E.K.); 2Department of Psychiatry and Psychotherapy, University Leipzig Medical Center, Leipzig University, 04103 Leipzig, Germany; 3Centre for Research, Further Education and Consulting, University of Applied Sciences for Social Work, Education and Nursing, 01307 Dresden, Germany; jan.schuhr@ehs-dresden.de (J.S.); tanja.brock@ehs-dresden.de (T.B.)

**Keywords:** COVID-19, university students, mental health, help-seeking, mental health service use, suicidal ideation

## Abstract

Background: Current evidence suggests that a significant proportion of university students are affected by mental disorders and suicidal ideation. Despite this, a treatment gap exists. Therefore, the present study assessed students’ knowledge and past use of on- and off-campus mental health services and help-seeking intentions. Furthermore, resilience was investigated as a potential barrier to help-seeking behavior. Methods: Data were collected between April and May 2022 from *N* = 5510 students from Saxony, Germany. To compare dependent variables, subgroups were computed according to students’ mental health status. Variables were assessed using standardized questionnaires. Chi-square tests were used for comparisons between groups. Multiple regression models were used to investigate the influence of resilience on help-seeking behavior. Results: Between 34 and 38% (depending on the subgroup) of participants indicated that they were not aware of their universities’ psychosocial counseling services. Furthermore, between 17 and 19% of participants indicated that they were not willing to seek help from professional mental health services. Finally, the previously found negative effect of resilience on help-seeking behavior was confirmed. Conclusion: The results showed a lack of awareness regarding universities’ mental health services and a treatment gap among university students. Universities and healthcare providers need to educate students about mental health services and how to access them. Further research is needed to elucidate the differential impact of resilience on mental health and help-seeking.

## 1. Introduction

Past studies indicate that university students constitute a vulnerable population, with up to one in three students suffering from a mental disorder globally [1,2]. In Germany, a 2018 report indicated that even before the SARS-CoV-2/COVID-19 pandemic, around 17% of students suffered from a diagnosed mental health condition [3], of which roughly 5% were apportioned to affective disorders. More recently, and in the context of the COVID-19 pandemic, a number of meta-analyses alleged a prevalence of depressive and anxious symptomatology of 34–63% and 31–55%, respectively, in university students [4,5,6]. A recent study conducted in Germany suggested that up to one in three students indicated suffering from moderate-to-severe symptoms of depression and more than one in four reported suffering from moderate-to-severe symptoms of anxiety [7]. Similarly, a recent study by our research group [8] found that more than one third of participants reported suffering from potentially clinically relevant symptoms of depression, anxiety, and/or alcohol misuse. Furthermore, we found that almost one in five students reported suicidal ideation (SI) or non-suicidal self-injury (NSSI), a marked increase in the prevalence of SI and NSSI among university students from 14.5% in 2020 to 19.6% in 2022 [8].

On principle, virtually all university students in Germany have access to professional treatment options thanks to universal healthcare. Additionally, most universities offer counseling services for students’ mental health [9]. These services’ primary goal is the prevention of the development and consolidation of mental health issues by providing low-threshold, short-term, age- and target-population-appropriate support [10,11,12]. Additionally, they also offer support in acute crises and in finding appropriate treatment options in the healthcare system, if long-term support or therapy is indicated.

However, available studies suggest that treatment rates for mental disorders among university students remain low, with studies indicating that only 16–30% of students in need receive treatment in a given year [1,13]. In the context of the COVID-19 pandemic, a recent German study found that only 6.5% of participants who reported moderate-to-severe levels of anxiety or depression had received treatment [7]. As suggested by previous studies, one reason might be that students lack knowledge about the counseling and treatment options available to them [14,15]. To our knowledge, however, there are no data on the level of awareness regarding university counseling services among students in Germany or their preferred treatment options. Further still, little is known about differences between students based on their mental health status. Such differences could inform stakeholders about special needs depending on the student’s mental health status.

Taking the low treatment rates among students into consideration, resilience was investigated as a predictor for help-seeking behavior (i.e., being currently in treatment). Special interest was placed on resilience, as previous studies have seldom investigated such personality variables as predictors for help-seeking behavior in the past. Resilience is commonly described as the ability to quickly recover, or “bounce back”, from stress and negative life events [16,17], and metanalytic evidence suggests a negative association with negative mental health indicators and a positive association with positive mental health indictors [18]. However, in one longitudinal cohort study, lower scores of resilience were found to be associated with increased levels of help-seeking in individuals diagnosed with major depressive disorder [19]. This result has yet to be replicated.

As such, the current study aimed to determine: (a) what proportion of university students are aware of the available mental health services; (b) what proportion of university students show help-seeking intentions and behaviors; (c) whether significant differences exist in help-seeking intentions and behaviors depending on students’ mental health status; and, finally, (d) whether resilience is a negative predictor of help-seeking behavior. Specifically, we assumed that higher levels of resilience would be associated with lower levels of help-seeking behavior.

## 2. Materials and Methods

### 2.1. Participants and Procedure

Details on the recruitment process are outlined in Kohls et al., 2023 [8]. The current study recruited a sample of students affiliated with six universities in Saxony, Germany, from 11 April to 27 May 2022, via an online survey. To accommodate international students and non-native German speakers, the survey could be completed in the German or English language.

For inclusion in the study, participants had to be 18 years or older, be enrolled at one of the participating universities, and provide informed consent. There were no other criteria for inclusion or exclusion. Approval for the study was granted by the ethics committee of Leipzig University on 11 November 2021 (file reference: 509/21-ek).

In total, *N* = 5510 participants completed the survey. After data cleaning, *n* = 36 participants had to be excluded from the final analysis, either because they indicated in a control question that they had not answered the survey conscientiously, they were underage, or their data were considered to be implausible during a manual check. As such, the final sample used in all following analyses contained *n* = 5474 cases.

In the current study, the sample was split into three subgroups (see Section 2.6) according to the self-reported mental health status (No Symptoms Group: below cut-offs, Symptoms Group: above cut-offs) and suicidal ideation (SI group) to compare awareness regarding mental health services and help-seeking intentions and behavior between these subgroups, as well as to investigate predictors of help-seeking behavior.

### 2.2. Measures

#### Sociodemographic Information

Information on age; gender (female, diverse, male); family status (single, in a relationship); history of migration (self, only parents, none); current study program (bachelor’s degree, master’s degree, other); monthly net income (EUR 1–499, EUR 500–1000, >EUR 1000, not specified); presence of (chronic) somatic conditions (yes, no); and parental educational attainment (low, middle, elevated, high) was collected (Table 1). To classify parental educational attainment into one of the aforementioned categories, participants were asked to name the highest educational qualification of both parents [20]. If both parents held a university degree, parental educational attainment was considered to be high. Educational attainment was considered to be elevated if only one parent held a university degree. Medium attainment was considered if both parents, and low if only one or none of the parents, had completed an apprenticeship, skilled worker’s qualification, master craftsman’s examination, technical school, or technician’s qualification. Taking the COVID-19 pandemic into account, current vaccination status was also assessed.

### 2.3. Mental Health Measures

To evaluate depressive symptoms, the Patient Health Questionnaire 9 (PHQ-9) was used [21]. The questionnaire consists of 9 items, and each item is rated on a 4-point rating scale (0 = “not at all”, 3 = “nearly every day”). A sum score (range: 0–27) is computed, with higher scores indicating a greater depressive symptom load. A score >9 is considered to be indicative of moderate depressive symptoms [21,22]. Additionally, the current study used item 9 (“Thoughts that you would be better off dead or of hurting yourself in some way”) of the PHQ-9 as an indicator for suicidal ideation. Any rating >0 on this item was considered to indicate suicidal ideation. Cronbach’s α was 0.85 (English) and 0.91 (German).

Symptoms of generalized anxiety disorder were assessed using the Generalized Anxiety Disorder scale (GAD-7) [23]. Seven items with a 4-point rating scale (0 = “not at all” to 4 = “nearly every day”) assess the severity of symptoms of generalized anxiety disorder across the past two weeks. Item scores are summed up to a total score, with higher scores indicating greater symptom severity. Likely cases of generalized anxiety disorder are identified by a sum score >10 [23,24]. Cronbach’s α was 0.91 (English) and 0.87 (German).

Alcohol misuse was evaluated utilizing the Alcohol Use Disorder Identification Test (AUDIT-C) [25,26]. This test encompasses three items, rated on a 5-point scale; a total score >4 identifies alcohol misuse in women, while a total score >5 identifies alcohol misuse in men. In the current study, the cut-off for women was used to identify alcohol misuse in self-identified diverse-gender participants. Cronbach’s α was 0.76 for both language versions of the survey.

Additionally, participants were asked via a single-choice item if they were currently in treatment for any mental disorder. Participants could indicate that they were receiving medication, psychotherapy, both, or neither.

### 2.4. Help-Seeking Intentions and Awareness and Use of Mental Health Resources

Awareness about the availability of on-campus mental health services was assessed utilizing two adapted items previously used in [27]. Students were asked if their university offered psychosocial counseling or online services as counseling and support services for students. Response options were “yes”, “no”, and “I don’t know”. Additionally, participants were asked if they had used each of the services before. Participants could indicate that they had never used the service, used it at least once in their lifetime, and/or used it more than once. The last two categories were combined into one for statistical analyses.

To assess help-seeking intentions in relation to off-campus mental health services, a list with common mental health services in Germany was developed through expert opinion and discussion in our research group. Participants were given this list of mental health services (psychiatrist, psychotherapy, psychological/psychosocial counseling, inpatient treatment in a psychiatric/psychosomatic clinic, online services for mental health problems) and were asked to rate on a 4-point scale (0 = “very unlikely”, 5 = “very likely”) how likely they were to use each service if they were experiencing psychological problems. Subsequently, for each service, participants were asked if they had used it during the past year, during their lifetime, or never. The first two categories were combined into one for statistical analyses. To gain further insight into help-seeking intentions, an additional variable was computed to identify participants’ unfavorable help-seeking intentions in relation to psychiatric and psychotherapeutic treatment, as well as psychosocial counseling.

### 2.5. Emotional and Personality Variables

In the present study, the Perceived Stress Scale (PSS-4) was used to evaluate participants’ stress [28]. The instrument consists of four items, rated on a 5-point scale (0 = “never”, 4 = “very often”). A total sum score can be computed, with higher scores indicating more perceived stress. Cronbach’s α was 0.80 (German language version) and 0.79 (English language version).

Resilience was assessed using the Brief Resilience Scale (BRS) [16]. The scale consists of six items that are rated on a 5-point Likert scale (1 = “strongly disagree” to 5 = “strongly agree”); the scale mean is used to quantify resilience, with higher scores indicating higher levels of resilience. Cronbach’s α was 0.77 (German language version) and 0.82 (English language version).

The UCLA 3-item Loneliness Scale (UCLA 3) was used to evaluate loneliness in the participants [29]. Using the three items, rated on a 3-point scale (1 = “hardly ever” to 3 = “often”), a total sum score is computed, with higher scores indicating more perceived loneliness. Cronbach’s α was 0.86 (German language version) and 0.80 (English language version).

Perceived emotional social support was estimated using the ENRICHD Social Support Inventory (ESSI) [30,31]. The five items are rated on a 5-point scale (1 = “none of the time” to 5 = “all of the time”) and combined to a sum score, with higher scores indicating higher perceived levels of social support. Cronbach’s α was 0.93 (German language version) and 0.88 (English language version).

### 2.6. Assignment to Subgroups

Participants were assigned to three different subgroups depending on their self-reported clinically relevant psychopathological symptoms. If participants had a score above the cut-off value for at least one of the instruments that were utilized in the present study (PHQ-9, GAD-7, or AUDIT-C; see also Section 2.3 “Mental Health Measures”), they were assigned to the Symptoms Group. If they indicated having had suicidal thoughts or thoughts of self-harm in item 9 of the PHQ-9, they were considered to be suffering from suicidal ideation. If a participant scored above or below a cut-off and reported suicidal ideation as well, they were assigned to the SI Group. If none of these criteria applied, participants were assigned to the No Symptoms Group.

### 2.7. Statistical Analysis

Statistical analyses were performed using IBM SPSS Statistics version 27.0. A two-tailed α = 0.05 significance level was used for statistical testing. Descriptive statistics for sociodemographic characteristics and outcomes regarding mental health, personality, social, and emotional variables were computed. The continuous age variable was transformed into distinct age groups in Table 1 for illustration purposes.

Differences in categorical variables between groups (sociodemographic variables and awareness and use of mental health resources) were investigated utilizing χ^2^ tests with 10,000 iteration bootstrapping. Significant differences were decomposed by comparing column proportions using the z test provided by SPSS. To estimate effect sizes, the φ coefficient was used, while Cramér’s V (φ_c_) was considered when the contingency table was larger than 2 × 2; φ, φ_c_ = 0.10 was interpreted as a small effect; φ, φ_c_ = 0.30 was interpreted as an average effect; and φ, φ_c_ = 0.50 was interpreted as a large effect.

Because of the non-normality of the dependent variables, the Kruskal–Wallis test procedure was used to investigate associations between group affiliation and help-seeking intentions regarding off-campus mental health resources. In the case of significance, the effects were decomposed further utilizing Dunn–Bonferroni tests. As no clear, interpretable pattern emerged, the score was dichotomized, with item scores <2 interpreted as unfavorable help-seeking intentions in the case of mental health problems, and scores >2 denoting positive help-seeking intentions. Differences in the newly computed categorical variable were analyzed using χ^2^ tests, as outlined above.

Three multivariable logistic regression models were computed to investigate predictors of currently receiving treatment in each separate subgroup. For this, the dependent variable ‘receiving treatment’ was dichotomized into either ‘currently receiving treatment’ (i.e., any of the treatment options were chosen as an answer) or ‘currently not receiving treatment’. Gender; age; relationship status; having children; current net income; parental educational attainment; migration status; being a foreign student; desired degree (bachelor’s degree, master’s degree, diploma, other); suffering from a chronic somatic disorder; loneliness; social support; stress; and resilience were added as predictor variables. As all VIF < 10, it was assumed that multicollinearity was not a confounding factor in any of the analyses. The explained variance was gauged using Nagelkerke’s pseudo *R^2^* [32].

## 3. Results

### 3.1. Sociodemographic Characteristics

The mean age in the current sample was 23.71 years (SD = 4.81), ranging from 18 to 56 years old. Concerning gender, *n* = 3775 (69.0%) reported being female, *n* = 1599 (29.2%) being male, and *n* = 100 (1.8%) being diverse. A detailed description of the total sample, containing *N* = 5474 participants, may be found in Kohls et al., 2023 [8]. Concerning the present study, a detailed description of the subgroups and results of statistical testing for subgroup differences is presented in Table 1. The No Symptoms Group encompassed a total of *n* = 1971 (36.0%), while *n* = 2428 (44.4%) participants were assigned to the Symptoms Group, and *n* = 1075 (19.6%) participants were assigned to the SI Group. An in-depth description of the aforementioned groups regarding mental health, personality, and emotional measures is available in the Appendix A).

**Table 1 behavsci-13-00885-t001:** Sociodemographic characteristics (*N* = 5474).

	No Symptoms (*n* = 1971)	Symptoms (*n* = 2428)	Suicidal Ideation (*n* = 1075)	*df*	χ^2^	φ_c_
Gender	(*n*, %)	(*n*, %)	(*n*, %)	4	51.34 ***	0.07
Female	1325 (67.2%) ^a^	1721 (70.9%) ^b^	729 (67.8%) ^a,b^			
Male	634 (32.2%) ^a^	660 (27.2%) ^b^	305 (28.4%) ^a,b^			
Diverse	12 (0.6%) ^a^	47 (1.9%) ^b^	41 (3.8%) ^c^			
Relationship status				2	29.27 ***	0.07
In a relationship	1013 (51.4%) ^a^	1209(49.8%) ^a^	446 (41.5%) ^b^			
Single	958 (48.6%) ^a^	1219 (50.2%) ^a^	629 (58.5%) ^b^			
Residential status				2	11.73 **	0.05
Alone	482 (24.5%) ^a^	565 (23.3%) ^a^	308 (28.7%) ^b^			
Being parent				2	39.95 ***	0.09
Yes	165 (8.4%) ^a^	113 (4.7%) ^b^	38 (3.5%) ^b^			
Migration status				4	25.49 ***	0.05
Self	108 (5.5%) ^a^	138 (5.7%) ^a^	92 (8.6%) ^b^			
Parents	110 (5.6%) ^a^	159 (6.5%) ^a,b^	93 (8.7%) ^b^			
No migration background	1753 (88.9%) ^a^	2131 (87.8%) ^a^	890 (82.8%) ^b^			
Parental education				6	31.85 ***	0.05
Low	56 (2.9%) ^a^	86 (3.5%) ^a,b^	49 (4.6%) ^b^			
Middle	682 (34.6%) ^a^	863 (355%) ^a^	348 (32.4%) ^a^			
Elevated	516 (26.2%) ^a^	601 (24.8%) ^a^	240 (22.3%) ^a^			
High	590 (29.9%) ^a^	760 (31.3%) ^a^	344 (32.0%) ^a^			
Do not know/ prefer not to say	127 (6.4%) ^a,b^	118 (4.9%) ^b^	94 (8.7%) ^a^			
Current income				8	19.81 *	0.04
No income	199 (10.1%) ^a,b^	227 (9.3%) ^b^	130 (12.1%) ^a^			
EUR 0–499/mo	369 (18.7%) ^a^	461 (18.9%) ^a^	218 (20.3%) ^a^			
EUR 500–999/mo	830 (42.1%) ^a^	1073 (44.2%) ^a^	480 (44.7%) ^a^			
≥EUR 1000/mo	510 (25.9%) ^a^	610 (25.1%) ^a^	223 (20.7%) ^b^			
Prefer not to say	63 (3.2%) ^a^	57 (23%) ^a^	24 (2.2%) ^a^			
Foreign student					12.03 **	0.05
Yes	82 (4.2%) ^a^	94 (3.9%) ^a^	69 (6.4%) ^b^			
Study program				4	45.58 ***	0.07
Bachelor’s	608 (30.8%) ^a^	867 (35.7%) ^b^	453 (42.1%) ^c^			
Master’s	1172 (59.5%) ^a^	1379 (56.8%) ^a^	557 (51.8%) ^b^			
Other	191 (9.7%) ^a^	182 (7.5%) ^b^	65 (6.0%) ^b^			
Somatic condition				2	27.33 ***	0.07
Yes	310 (15.7%) ^a^	382 (15.7%) ^a^	241 (22.4%) ^b^			
In treatment						
Yes	98 (5.0%) ^a^	241 (9.9%) ^b^	252 (23.5%) ^c^	2	249.82 ***	0.21
Age (*M, SD*)	23.95 (5.23)	23.69 (4.69)	23.30 (4.16)	2	5.08	

Notes: * *p* < 0.05, ** *p* < 0.01, *** *p* < 0.001; χ^2^, chi-square test statistic; φ_c_, Cramér’s *V*; calculation of % from valid cases in the same subgroup; ^a–c^, cells with the same superscript in a row did not differ significantly; “in treatment” refers to treatment for any mental disorder.

Significant associations between group affiliation and gender, relationship status, residential status, being a parent, migration status, current income, study program, suffering from a somatic condition, and being currently in treatment for any mental disorder were found. While significant, and with the exception of being currently in treatment for any mental disorder, all φ_c_ values were below >0.10. As such, the associations could be interpreted as being very small. Thus, a further decomposition of the identified effects was deemed unnecessary. Regarding being currently in treatment for any mental disorder, there were significant differences in treatment rates between all groups. Participants in the SI Group reported the highest treatment rates, followed by those in the Symptoms Group, while participants in the No Symptoms Group reported the lowest treatment rates. Furthermore, there were no differences regarding age between groups.

### 3.2. Awareness and Use of Mental Health Services at Universities

All in all, more than one third (37.0%) of students denied or did not know that there were psychological counseling services at their university (Table 2). A chi-square test for independence did not find a significant association between group affiliation and awareness about the availability of psychological counseling services for students at universities. A comparison of use rates revealed a small but significant association between group affiliation and the use of psychological counseling services for students at universities. The proportion of use was significantly different between all groups, with the largest use rate in the SI Group, followed by the Symptoms Group, and the lowest use rate in the No Symptoms Group.

Regarding online mental health services, within the No Symptoms Group, about half of the participants reported that they did not know if their institution offered these (Table 2). This proportion was even higher in the Symptoms Group and was highest in the SI Group. While a significant association between group affiliation and awareness about a university’s online mental health resources was found, the effect size was below the threshold to be considered small. Regarding differences in the use of online mental health services, no significant difference between groups emerged.

### 3.3. Help-Seeking Intentions and Use of Off-Campus Mental Health Services

The descriptive statistics of the continuous variable showed that help-seeking intentions regarding potential future use were most positive for psychotherapy (*M* = 2.97, *SD* = 1.07) and psychosocial counseling (*M* = 2.88, *SD* = 1.04), while inpatient treatment (*M* = 1.65, *SD* = 0.91) was viewed as the least favorable among the total sample. The results of testing for differences between groups with the dichotomized variable in terms of help-seeking intentions regarding off-campus mental health services are presented in Table 3.

Help-seeking intentions regarding the use of a psychiatrist’s services only differed significantly between the No Symptoms and SI Groups, with a greater proportion of participants in the SI Group endorsing their use, while no differences were found between the No Symptoms and Symptoms Groups or between the Symptoms and SI Groups. Regarding help-seeking intentions for psychotherapy, no significant differences were found between subgroups. Significant differences between subgroups were found for help-seeking intentions regarding psychosocial counseling between the SI and No Symptoms Groups, with a greater proportion of participants in the No Symptoms Group endorsing its use, as well as between the SI and Symptoms Groups, with a greater proportion of participants in the Symptoms Group endorsing its use. However, there was no significant difference when comparing help-seeking intentions between the No Symptoms and Symptoms Groups. Concerning help-seeking intentions in relation to inpatient treatment, significant differences between the Symptoms and SI Groups and No Symptoms and SI Groups were found, with a greater proportion of participants in the SI Group endorsing its use. A comparison of the No Symptoms and Symptoms Groups did not return significant results. Finally, help-seeking intentions regarding the use of online resources did not differ significantly between subgroups.

Focusing on participants who reported unfavorable help-seeking intentions for psychiatric and psychological treatment as well as psychosocial counseling, the proportions were 18.4% (*n* = 363; No Symptoms Group); 16.5% (*n* = 400; Symptoms Group); and 19.3% (*n* = 207; SI Group). There were no significant differences between groups, *χ*^2^_(2)_ = 4.98, *p* = 0.086.

When comparing the past use of those reporting clinically relevant psychopathological symptoms, significant differences were found for all comparisons. The results consistently pointed towards more past use among the Symptoms and SI Groups, in that order (Table 4). An exception was inpatient treatment in a psychiatric/psychosomatic clinic, for which only the use rate in the SI Group was significantly higher than in the other groups. The effect sizes were small for all comparisons.

Finally, significant differences were found between all groups regarding current treatment, with the SI Group having the greatest proportion of participants currently in treatment, followed by the Symptoms Group and, with a smaller proportion still, the No Symptoms Group.

### 3.4. Predictors for Current Treatment

For the SI Group, the overall model was significant, but the proportion of explained variance was small (Table 5). The only significant positive predictor for being currently in treatment was identifying as diverse gender. Higher resilience was a negative predictor, as was being enrolled in a master’s program.

Regarding the Symptoms Group, the overall model was significant (Table 6). The proportion of explained variance was small. Older age was the only significant positive predictor for help-seeking behavior, while suffering from a somatic condition, being enrolled in a master’s program, and higher resilience were associated with a smaller likelihood of currently being in treatment.

Finally, among the No Symptoms Group, the resulting model was again significant, and the proportion of explained variance was small (Table 7). Diverse gender, older age, and higher loneliness positively predicted help-seeking behavior, while suffering from a somatic condition and higher resilience were negative predictors for being currently in treatment.

## 4. Discussion

The present study is among the first to assess knowledge, help-seeking intentions, and the use of on- and off-campus mental health resources among university students in Germany, with regard to students’ mental health status. For this, the sample was split into three subgroups to examine differences between participants without symptoms; participants currently suffering from depressive, anxious, or alcohol misuse symptoms; and participants suffering from SI. Furthermore, resilience and sociodemographic variables were examined as predictors of current help-seeking behavior in each of the subgroups. First and foremost, the current study provides further support for the argument that a considerable proportion of students are poorly informed about the mental health resources offered by their institutions. Help-seeking intentions, as well as past use, differed according to mental health status, although effect sizes were very small. A considerable proportion of students indicated they were not willing to seek help from mental health services. Furthermore, the previously identified negative effect of resilience on help-seeking could be confirmed and extended to actual mental health help-seeking behavior in students in Germany.

### 4.1. Awareness and Use of Mental Health Services at Universities

The descriptive results indicated that more than one third of the participants were unaware of the psychosocial counseling services at their universities, and more than half stated that they did not know if their university offered any online mental health resources. This result, indicating limited knowledge about universities’ mental health resources among some students, is in line with other studies conducted in Germany [14,15]. Further, this phenomenon is not limited to Germany (a cursory literature search uncovered evidence for this knowledge gap in students in Ethiopia [33], Canada [34], Northern Ireland [35], and the US [36]), nor does it seem to be a new development [37].

The differences in awareness about the available services showed that individuals suffering from SI were better informed about the psychosocial counseling services at their universities but less aware of online mental health resources. Still, more than one third of those suffering from SI stated that they were not aware of denied the availability of counseling services at their university. As such, there is a need for more effective communication regarding available services and outreach to students in need. Furthermore, there is some evidence that individuals suffering from SI are more likely to prefer online settings to face-to-face settings [38,39]. As they are low-cost and low-threshold, making online services available and raising awareness about their availability seems important in the context of students suffering from SI. While the current study did not identify differences in the past use of online mental health resources between groups, and with only up to 13% of students reporting the use of online mental health resources provided by their university in the past, the item assessing past use did not differentiate between the use of online resources such as websites and digitally mediated counseling or therapy; thus, no statements can be provided about the type of online resources used.

Taking the past use of these services into account, about one in five students suffering from SI, one in ten students currently suffering from relevant psychopathological symptoms, and roughly one in twenty students without current symptoms had used a university’s psychosocial counseling service before.

### 4.2. Help-Seeking Intentions and Use of Off-Campus Mental Health Services

While significant differences in help-seeking intentions between the subgroups were found, the effect sizes were very small. Thus, drawing inferences from the present results seemed unwarranted. However, the descriptive results showed generally favorable help-seeking intentions regarding psychotherapeutic and psychiatric treatment as well as psychosocial counseling. Most participants not suffering from assessed symptoms (69%), as well as those suffering from symptoms (71%) and those suffering from SI (72%), indicated that they were likely to seek psychotherapeutic and psychiatric treatment (No Symptoms Group, 49%; Symptoms Group, 50%; SI Group, 54%) or psychosocial counseling (No Symptoms Group, 69%; Symptoms Group, 70%; SI Group, 63%) in the case of mental health problems. This general endorsement was somewhat in contrast to past use and current treatment rates among the SI and Symptoms Groups. All in all, the current study provides evidence of a slight increase compared to previously reported help-seeking intentions and use rates [40]. However, as the method of assessment for treatment need and mental health help-seeking behavior differs between studies, comparability remains limited. Furthermore, even among those suffering from SI, more than half never sought treatment, which is a higher rate than some previous evidence for US students suggests [41,42].

Additionally, depending on mental health status, in the current study, between 17 and 19% of participants indicated that they were unlikely to seek help from any mental health professional (psychiatric and psychotherapeutic treatment or psychosocial counseling). While this may be interpreted as a help-negation effect, previously defined as the help-avoidance of those suffering from SI or psychological distress [43,44], in the current study no differences between the No Symptoms, Symptoms, and SI Groups could be identified. As such, the current results suggest no specific help-negation effect in the present sample. Help-seeking avoidance is commonly associated with a lack of mental health literacy and (self-)stigma [45]. Further, male gender, non-white ethnicity, low socioeconomical status, and being a first-generation student were also found to negatively influence the likelihood of seeking university mental health services, whereas for gender, the results were inconsistent [46,47].

Regarding online help services for mental health problems, the proportion of those likely to seek help via this modality only differed significantly between the SI (36%) and Symptoms Groups (40%), with significantly fewer participants in the SI Group endorsing help-seeking from online help services. However, when asked about past use, the proportion was highest among the SI Group (28%), differing significantly from the No Symptoms (14%) and Symptoms Groups (20%). Further research is needed to elucidate potential reasons for this seemingly counterintuitive result, given some past evidence supporting a preference for online settings among those suffering from SI [38,39]. Also, given the relatively low rates of help-seeking intentions via online modalities in the current study, research on ways to increase the acceptance of online mental health services is needed to fully realize their potential. Tendencies are inherently linked to the various types of resources accessible through online mental health services. Considering support seeking and information seeking [48], the latter has been observed to be prevalent among university students, with an average occurrence of around one third, though its prevalence spans a wide range from 11.8% to 92.4% across various studies [49]. Students attribute several advantages to seeking information online, including the constant availability and cost-free access to a diverse range of information, along with the option of maintaining anonymity [50]. Conversely, common disadvantages include challenges in sifting through the vast amount of available data to identify accurate and relevant information, as well as uncertainties about its quality and reliability [49]. Furthermore, the attributes of anonymity, accessibility, and cost-effectiveness play a pivotal role in the act of seeking support through e-mental health services. Moreover, these services offer the potential to shape discussions on mental health matters and foster a sense of solidarity among peers [51]. However, concerns related to the online format revolve around issues of data security, the effectiveness of digital tools, and the difficulty of maintaining genuine and intimate conversations within the confines of digital communication platforms [49,52].

### 4.3. Resilience as Predictor for Help-Seeking Behavior

In line with previous evidence [19], resilience negatively predicted help-seeking behavior among all subgroups. With only limited literature to draw upon, the precise nature of the relationship between resilience and help-seeking remains unclear. Mirroring the rationalization given by past researchers, it may be that resilience is protective against the occurrence of mental disorders [16,17] but at the same time acts as a barrier for help-seeking, because resilience might be associated with the wish the individual to deal with problems autonomously [19]. Further longitudinal research seems warranted to disentangle this effect.

### 4.4. Implications for Stakeholders

In a 2021 report, it was noted that roughly 38% of German students reported feeling in need of counseling because of a depressive mood [53]. However, only 26% of those students reported using a counseling service. Still, service providers have reported a significant increase in contact inquiries over time, and there are some indications that at least some of those providers are already operating at capacity, e.g., one provider reported waiting periods of up to twelve weeks in 2021 [53], while another reported having to refer 13% of contact requests to other mental health professionals due to being at maximum capacity in 2022 [54]. With past reports of an increase in the prevalence of mental disorders in the student population over time [3,55] and a potentially sustained negative impact of the pandemic on the mental health of university students [8,56], what measures can be taken to increase help-seeking behavior among university students in Germany?

Cross-national studies have suggested different student and service qualities that influence the likelihood of usage, with greater availability being an overall key indicator for greater use [46]. As such, increasing availability, an important objective in and of itself, may also increase help-seeking behavior among university students. As mentioned above, reports from service providers suggest limited availability as well as a steady increase in demand over time [12,54]. This development is leading to prolonged waiting periods for an initial consultation, thus decreasing availability [57]. The Deutsches Studierendenwerk (an association concerned with, inter alia, social and health support for students at German universities) recently cautioned political stakeholders against this development and encouraged them to take action and grant resources to expand counseling services at universities in order to increase their availability for students in need. Therefore, it is imperative that universities’ counseling service providers are sufficiently funded and staffed. At the local level, interventions improving help-seeking behavior should be considered. Evidence for the efficacy of interventions targeting mental health literacy, stigma, and help-seeking behavior exists [58], and the implementation of such interventions at universities is advised. Furthermore, the provision of low-threshold online services and engaging and easily accessible online resources may supplement face-to-face, in-person counseling and improve help-seeking behavior [59,60]. However, as noted above, recent literature and the present findings suggest the limited use of online and digital interventions by the student body. Therefore, further research is needed to clarify the conditions necessary to facilitate use by those in need. Regarding minority students, needs and conditions are to be met with adequate knowledge (e.g., cultural background) concerning distinctive causes to avoid additional stigmatization. That being said, all this presupposes adequate funding, which needs to be provided by political actors and institutions to enable service providers to expand and transform their services.

### 4.5. Strengths and Limitations

The current study could take advantage of a large sample, collected at a range of different universities in Saxony, Germany. Some limitations of the current study should be taken into account. First and foremost, the sample was collected in the immediate aftermath of social distancing measures and lockdowns (because of the COVID-19 pandemic in Germany), and it is unclear if there have been additional changes, beyond those previously reported [8], to the mental health status of students since then. As the sample was a convenience sample, its generalizability may be limited due to students with mental disorders being more likely to self-select for participation in the study as well as the underrepresentation of some sociodemographic groups (e.g., men, bachelor’s program students). Furthermore, the current study relied on self-reporting instruments, and, due to the cross-sectional nature of this study, no causal inferences could be drawn.

## 5. Conclusions

Most importantly, the results imply a lack of awareness among students regarding mental health services at universities and, at the same time, show that only a fraction of those in need are able or willing to use any on- or off-campus professional mental health services. Furthermore, across all investigated groups, about one fifth of the participants indicated that it was unlikely for them to seek help from professional mental health services. As such, universities and mental healthcare providers need to educate university students about mental health services and how to access them and promote the timely and preventive use of low-threshold mental health services. Furthermore, interventions addressing common barriers to help-seeking behaviors, such as mental health literacy and stigma, should be implemented, e.g., as part of the curriculum of university students, to address the existing treatment gap. Further still, political stakeholders need to provide adequate funding to ensure the sufficient availability and staffing of services.

## Figures and Tables

**Table 2 behavsci-13-00885-t002:** Knowledge and past use of psychosocial counseling and online services at universities (*N* = 5474).

		No Symptoms	Symptoms	Suicidal Ideation	*df*	*χ^2^*	*φ_c_*
Knowledge		(*n*, %)	(*n*, %)	(*n*, %)			
Psychosocial counseling for students					4	9.17	0.03
	Yes	1239 (62.9%)	1510 (62.2%)	702 (65.8%)			
	No	28 (1.5%)	46 (1.8%)	28 (2.6%)			
	Do not know	704 (35.7%)	872 (35.9%)	345 (31.6%)			
Online services					4	27.55 ***	0.05
	Yes	982 (49.8%) ^a^	1106 (45.6%) ^b^	433 (40.3%) ^c^			
	No	56 (2.8%) ^a^	81 (3.3%) ^a^	46 (4.3%) ^a^			
	Do not know	933 (47.7%) ^a^	1241 (50.9%) ^b^	596 (55.4%) ^b^			
Past use							
Psychosocial counseling for students		135 (6.8%) ^a^	274 (11.3%) ^b^	213 (19.8%) ^c^	2	116.11 ***	0.15
Online services		253 (12.8%)	298 (12.3%)	145 (13.5%)	2	1.03	

Notes: *** *p* < 0.001; ^a–c^, cells with the same superscript in a row did not differ significantly.

**Table 3 behavsci-13-00885-t003:** Help-seeking intentions regarding mental health services.

	No Symptoms	Symptoms	Suicidal Ideation	*df*	χ^2^	φ_c_
Would use	(*n*, %)	(*n*, %)	(*n*, %)			
Psychiatrist	965 (49.0%) ^a^	1229 (50.6%) ^a,b^	581 (54.0%) ^b^	2	7.21 *	0.04
Psychotherapy	1350 (68.5%)	1733 (71.4%)	771 (71.7%)	2	5.45	0.03
Psychological/psychosocial counseling	1369 (69.5%) ^a^	1703 (70.1%) ^a^	679 (63.2%) ^b^	2	18.06 ***	0.06
Inpatient treatment in a psychiatric/psychosomatic clinic	320 (16.12%) ^a^	377 (15.5%) ^a^	275 (25.6%) ^b^	2	56.46 ***	0.10
Online help services for mental health problems	786 (39.9%) ^a,b^	979 (40.3%) ^b^	385 (35.8%) ^a^	2	6.81 *	0.04

Notes: * *p* < 0.05, *** *p* < 0.001; ^a,b^, cells with the same superscript in a row did not differ significantly.

**Table 4 behavsci-13-00885-t004:** Past and current use of mental health services.

	No Symptoms	Symptoms	Suicidal Ideation	*df*	*χ* ^2^	*φ_c_*
Used before	(*n*, %)	(*n*, %)	(*n*, %)			
Psychiatrist	161 (8.2%) ^a^	294 (12.1%) ^b^	244 (22.7%) ^c^	2	133.54 ***	0.16
Psychotherapy	328 (16.6%) ^a^	626 (25.8%) ^b^	452 (42.0%) ^c^	2	235.23 ***	0.21
Psychological/psychosocial counseling	248 (12.6%) ^a^	463 (19.1%) ^b^	333 (31.0%) ^c^	2	152.49 ***	0.17
Inpatient treatment in a psychiatric/ psychosomatic clinic	79 (4.0%) ^a^	120 (4.9%) ^a^	123 (11.4%) ^c^	2	76.40 ***	0.12
Online help services for mental health problems	265 (13.4%) ^a^	483 (19.9%) ^b^	301 (28.0%) ^c^	2	96.63 ***	0.13
Currently in treatment						
Yes	98 (5.0%) ^a^	241 (9.9%) ^b^	252 (23.4%) ^c^	2	249.57 ***	0.21

Notes: *** *p* < 0.001; χ^2^, chi-square test statistic; φ, phi coefficient; calculation of % from valid cases of the same subgroup; ^a–c^, cells with the same superscript in a row did not differ significantly.

**Table 5 behavsci-13-00885-t005:** Predictors for current treatment among SI Group (*n* = 1075).

	B	SE	*p*	*OR*	*OR 95% CI*	
Gender (ref.: female)						
Male	−0.22	0.19	0.252	0.80	0.55	1.17
Diverse	1.20	0.35	0.001 **	3.32	1.67	6.58
Relationship status (ref.: in a relationship)						
Single	−0.06	0.17	0.743	0.94	0.67	1.33
Residential status (ref.: alone)						
Shared	−0.12	0.17	0.496	0.89	0.63	1.25
Being parent (ref.: yes)						
No	0.61	0.49	0.219	1.84	0.70	4.84
Migration status (ref.: self)						
Parents	0.04	0.39	0.922	1.04	0.48	2.24
No migration background	0.09	0.31	0.761	1.10	0.60	2.02
Parental education (ref.: low)						
Middle	0.21	0.42	0.604	1.24	0.55	2.80
Elevated	0.17	0.43	0.690	1.18	0.51	2.73
High	0.54	0.41	0.191	1.71	0.77	3.82
Prefer not to say	0.44	0.46	0.337	1.55	0.63	3.78
Current income (ref.: no income)						
EUR 0–499/mo	−0.09	0.29	0.767	0.92	0.52	1.63
EUR 500–999/mo	0.36	0.25	0.159	1.43	0.87	2.35
≥EUR 1000/mo	0.46	0.29	0.108	1.59	0.90	2.81
Prefer not to say	0.30	0.54	0.582	1.35	0.47	3.89
Foreign student (ref.: yes)						
No	0.06	0.36	0.863	1.06	0.53	2.15
Study program (ref.: bachelor’s)						
Master’s	−0.32	0.16	0.049 *	0.73	0.53	1.00
Other	−0.31	0.34	0.361	0.73	0.37	1.43
Somatic condition (ref.: yes)						
No	−0.24	0.18	0.169	0.78	0.55	1.11
Age	0.04	0.02	0.036	1.05	1.00	1.09
Loneliness (UCLA)	0.02	0.05	0.692	1.02	0.93	1.12
Social support (ESSI)	0.03	0.02	0.147	1.03	0.99	1.07
Stress (PSS-4)	−0.01	0.03	0.670	0.99	0.93	1.05
Resilience (BRS)	−0.75	0.13	0.000 ***	0.47	0.37	0.61
Constant	−1.68	1.30	0.196	0.19		
*χ* ^2^		χ^2^_(24)_ = 94.50, *p* < 0.001				
*R*^2^ (*Nagelkerke*)		0.13				

Notes: * *p* < 0.05, ** *p* < 0.01, *** *p* < 0.001; OR, odds ratio; CI, confidence interval; UCLA 3, Three-item Loneliness Scale; ESSI, ENRICHD Social Support Inventory; PSS-4, Perceived Stress Scale; BRS, Brief Resilience Scale.

**Table 6 behavsci-13-00885-t006:** Predictors for current treatment among Symptoms Group (*n* = 2428).

	B	SE	*p*	*OR*	*OR 95% CI*	
Gender (ref.: female)						
Male	−0.17	0.19	0.364	0.84	0.58	1.22
Diverse	0.61	0.37	0.099	1.85	0.89	3.83
Relationship status (ref.: in a relationship)						
Single	0.22	0.16	0.172	1.25	0.91	1.73
Residential status (ref.: alone)						
Shared	−0.08	0.17	0.652	0.92	0.66	1.30
Being parent (ref.: yes)						
No	0.45	0.39	0.254	1.57	0.72	3.40
Migration status (ref.: self)						
Parents	−0.19	0.40	0.638	0.83	0.38	1.82
No migration background	−0.42	0.32	0.187	0.66	0.35	1.23
Parental education (ref.: low)						
Middle	0.34	0.45	0.446	1.41	0.59	3.37
Elevated	0.64	0.45	0.152	1.89	0.79	4.53
High	0.48	0.45	0.281	1.62	0.68	3.87
Prefer not to say	0.23	0.53	0.661	1.26	0.45	3.52
Current income (ref.: no income)						
EUR 0–499/mo	0.37	0.31	0.238	1.45	0.78	2.67
EUR 500–999/mo	0.50	0.29	0.081	1.65	0.94	2.91
≥EUR 1000/mo	0.11	0.32	0.730	1.12	0.60	2.10
Prefer not to say	−0.71	0.70	0.306	0.49	0.12	1.92
Study program (ref.: bachelor’s)						
Master’s	−0.36	0.15	0.020 *	0.70	0.52	0.95
Other	−0.31	0.31	0.309	0.73	0.40	1.34
Foreign student (ref.: yes)						
No	−0.36	0.42	0.399	0.70	0.30	1.61
Somatic condition (ref.: yes)						
No	−0.38	0.18	0.030 *	0.68	0.48	0.96
Age	0.08	0.02	0.000 ***	1.08	1.05	1.12
Loneliness (UCLA)	0.03	0.05	0.517	1.03	0.94	1.13
Social support (ESSI)	−0.02	0.02	0.392	0.98	0.94	1.02
Stress (PSS-4)	0.00	0.03	0.874	1.00	0.95	1.07
Resilience (BRS)	−1.31	0.13	0.000 ***	0.27	0.21	0.35
Constant	−0.76	1.27	0.547	0.47		
*χ* ^2^		*χ*^2^_(24)_ = 253.30, *p* < 0.001				
*R*^2^ (*Nagelkerke*)		0.21				

Notes: * *p* < 0.05, *** *p* < 0.001; OR, odds ratio; CI, confidence interval; UCLA 3, Three-item Loneliness Scale; ESSI, ENRICHD Social Support Inventory; PSS-4, Perceived Stress Scale; BRS, Brief Resilience Scale.

**Table 7 behavsci-13-00885-t007:** Predictors for current treatment among No Symptoms Group (*n* = 1971).

	B	SE	*p*	*OR*	*OR 95% CI*	
Gender (ref.: female)						
Male	0.18	0.27	0.497	1.20	0.71	2.02
Diverse	2.32	0.71	0.001 **	10.20	2.56	40.67
Relationship status (ref.: in a relationship)						
Single	−0.32	0.26	0.212	0.73	0.44	1.20
Residential status (ref.: alone)						
Shared	−0.27	0.27	0.306	0.76	0.45	1.28
Being parent (ref.: yes)						
No	−0.10	0.45	0.832	0.91	0.37	2.21
Migration status (ref.: self)						
Parents	−0.37	0.80	0.640	0.69	0.14	3.29
No migration background	0.02	0.53	0.977	1.02	0.36	2.89
Parental education (ref.: low)						
Middle	1.22	1.05	0.245	3.40	0.43	26.81
Elevated	1.20	1.06	0.257	3.32	0.42	26.45
High	1.44	1.06	0.171	4.23	0.54	33.48
Current income (ref.: no income)						
EUR 0–499/mo	0.79	0.55	0.148	2.21	0.75	6.46
EUR 500–999/mo	0.74	0.51	0.149	2.09	0.77	5.72
≥EUR 1000/mo	0.63	0.54	0.244	1.87	0.65	5.38
Prefer not to say	0.51	0.89	0.564	1.67	0.29	9.49
Study program (ref.: bachelor’s)						
Master’s	−0.23	0.24	0.349	0.80	0.50	1.28
Other	−0.20	0.42	0.631	0.82	0.36	1.86
Somatic condition (ref.: yes)						
No	−0.80	0.25	0.001 **	0.45	0.27	0.73
Age	0.06	0.02	0.008 **	1.06	1.02	1.12
Loneliness (UCLA 3)	0.16	0.07	0.034	1.17	1.01	1.35
Social support (ESSI)	−0.01	0.04	0.721	0.99	0.92	1.06
Stress (PSS-4)	−0.04	0.05	0.429	0.96	0.87	1.06
Resilience (BRS)	−1.40	0.19	0.000 ***	0.25	0.17	0.36
Constant	−1.31	2.08	0.529	0.27		
*χ* ^2^		*χ*^2^_(24)_ = 125.45, *p* < 0.001				
*R*^2^ (*Nagelkerke*)		0.19				

Notes: ** *p* < 0.01, *** *p* < 0.001; OR, odds ratio; CI, confidence interval; UCLA 3, Three-item Loneliness Scale; ESSI, ENRICHD Social Support Inventory; PSS-4, Perceived Stress Scale; BRS, Brief Resilience Scale.

## Data Availability

The raw data supporting the conclusions of this article will be made available by the authors upon request. Data are not publicly available due to privacy restrictions.

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
