# Peer review of "Differences in Help-Seeking Behavior among University Students during the COVID-19 Pandemic Depending on Mental Health Status: Results from a Cross-Sectional Survey"

_behavsci, 2023, doi:10.3390/bs13110885_

Round 1

Reviewer 1 Report

Comments and Suggestions for Authors

Thank you for inviting me to review this article, sorry for the wait. 

Please note that the manuscript is of high quality. 

Best regards. 

Reviewer 2 Report

Comments and Suggestions for Authors

The issue of the article is a current topic. In my opinion the manuscript has got a high quality: the scientific framework is wide, the logic of the analysis is clear, the limitations are correct. The number of the respondents are high enough and the applied methods are adequate. The results are practical and informative.

I have got only few suggestions:

-         the related data were represented in the theoretical frames but in my opinion the place and role of the mental health service at universities have to be more explained -  e.g. the history of it too.

-        the authors have to formulate research questions and/or hypotheses (I can see the aims but these are not the same).

-        in the case of subsection 2.2 the authors have to refer to Table 1

-        authors have to improve the formatting of the tables (especially the line spacing)

-        the results of the multiple regression models may be depicted in some way

-        two dots remain in the last row of 4.2.

-        I think the line spacing is not adequate in the reference list.

I suggest some improvement in the case of the theoretical framework, replacement of hypotheses or research questions and the depiction of the regression models in some way (perhaps in a table)

Reviewer 3 Report

Comments and Suggestions for Authors

·      Title seems missing a descriptor:

o   Differences in university students’ help-seeking XXXX (behavior? tendencies?) depends on mental health status:……”

·      Introduction:

o   This is way too long for an Introduction. It’s hard to follow because authors jumped from one point to another. Consider removing several paragraphs and include them in the Discussion instead. My recommendation is to keep the Introduction concise and formulate it with points such as: i) description of current state; ii) what are the gaps that we need to address; iii) objective(s) of the study. Other content can go to Discussion.

·      Materials and Methods:

o   “The current study takes advantage of a sample of students,...." should be revised to "The current study recruited a sample of students affiliated with six universities in Saxony, Germany, …."

o   Do you know how many total students received your survey? It's important to know the response rate. You mentioned that the sample size was 5,510 participants, but is this indicating at least 30% response rate? It is important to provide this indicator as a measure of study/data quality and representativeness.

·      Results:

o   I don’t see any summary table related to results of mental health measures (PHQ, GAD, AUDIT, treatment of mental health disorder; listed in section 2.3), health seeking intentions (listed in section 2.4), emotional/personality variables (PSS, UCLA, BRS, ESSI; listed in section 2.5) for the 3 groups (no symptoms, symptoms, SI). Variables such as mean, standard error or standard deviation, min- max range should be included as summary table to provide a more holistic description of the participants.

·      Discussion or Conclusion:

o   I understand that the key findings cannot be generalized to wider scale due to study limitations, however, it’ll be informative if authors could provide some in-depth suggestions on strategies that universities or mental health professionals in Germany could do to mitigate these issues. The manuscript has been very detailed in many ways but has glossed over this point, which seems to be a shortcoming that can be easily addressed to improve its value.

Comments on the Quality of English Language

Please note above comments on title and paraphrasing a sentence.
